# Wet Synthesis of Elongated Hexagonal ZnO Microstructures for Applications as Photo-Piezoelectric Catalysts

**DOI:** 10.3390/ma13132938

**Published:** 2020-06-30

**Authors:** Rosanna Pagano, Chiara Ingrosso, Gabriele Giancane, Ludovico Valli, Simona Bettini

**Affiliations:** 1Department of Biological and Environmental Sciences and Technologies, DISTEBA, University of Salento, Via per Arnesano, I-73100 Lecce, Italy; rosanna.pagano@unisalento.it (R.P.); ludovico.valli@unisalento.it (L.V.); 2CNR-IPCF Sez. Bari, c/o Department of Chemistry, University of Bari, via Orabona 4, I-70126 Bari, Italy; c.ingrosso@ba.ipcf.cnr.it; 3Department of Cultural Heritage, University of Salento, Via D. Birago, 48, I-73100 Lecce, Italy; 4Consorzio Interuniversitario Nazionale per la Scienza e Tecnologia dei Materiali, INSTM, Via G. Giusti, 9, I-50121 Firenze, Italy; simona.bettini@unisalento.it; 5Department of Engineering for Innovation, University of Salento, Via per Monteroni, I-73100 Lecce, Italy

**Keywords:** ZnO, piezopotential, photodegradation, UV-Vis spectroscopy, piezo-photodegradation

## Abstract

It is well known that energetic demand and environmental pollution are strictly connected; the side products of vehicle and industrial exhausts are considered extremely dangerous for both human and environmental health. In the last years, the possibility to simultaneously photo-degrade water dissolved pollutants by means of ZnO nanostructures and to use their piezoelectric features to enhance the photo-degradation process has been investigated. In the present contribution, an easy and low-cost wet approach to synthetize hexagonal elongated ZnO microstructures in the wurtzite phase was developed. ZnO performances as photo-catalysts, under UV-light irradiation, were confirmed on water dissolved methylene blue dye. Piezoelectric responses of the synthetized ZnO microstructures were evaluated, as well, by depositing them into films onto flexible substrates, and a home-made layout was developed, in order to stimulate the ZnO microstructures deposited on solid supports by means of mechanical stress and UV photons, simultaneously. A relevant increment of the photo-degradation efficiency was observed when the piezopotential was applied, proposing the present approach as a completely eco-friendly tool, able to use renewable energy sources to degrade water solved pollutants.

## 1. Introduction

Increasing energetic request has pushed many countries to invest in the development of technologies able to exploit energetic sources that are alternatives to fossil fuels [1,2]. Furthermore, the effects on human and environmental health induced by the waste products of the fossil fuels combustion are the cause of criticisms, such as air pollution and global warming. Photovoltaic, wind, hydroelectric and geothermic energy are widely used to supply energy for both domestic and industrial needs. Another class of energy harvesting devices, more devoted to supply energy for micro-power devices, comprises systems that need low power [3], such as watches, wearable wrists [4,5] or body sensors [6]. A particularly intriguing subset of this class of devices is represented by the devices which are able to obtain energy from the human body and daily life movements [7,8]. Among them, the devices based on the piezoelectric effect are attracting the interest of the scientific community, since they can be used not only for energy harvesting processes [9,10,11], but also in daily used devices, such as, for example, microphones and vibration sensors [12]. Direct and inverse piezoelectric effect are well-known phenomena since the end of the XIX century [13,14], which can be exploited in resonators [15], micropumps [16], or more complex systems like sonars [17], continuous pressure monitor devices, [18] and in the so-called piezoelectric surgery [19]. Alongside this, pushed by the dramatic increases of energy request, the use of piezoelectric materials, as devices for energy harvesting, appears to be a very appealing research field [20,21,22,23]. In fact, the possibility of miniaturizing the piezoelectric transductors allowed their integration in shoes to convert, at each step, the effect of the pressure in electric energy [24], or to put them near the eyes of quadriplegic patients in order to facilitate the communication [25].

The most studied and characterized materials able to produce a bias, as a consequence of a mechanical stress, are the PZT (lead zirconate titanate) [26], BaTiO_3_ (barium titanate) [27], quartz [28] and ZnO [29,30]. Zinc oxide, in the wurtzite crystalline phase, couples the piezoelectric features to the characteristics of catalysts for the water oxidation and degradation of organic pollutants under UV-Vis illumination [31,32]. In recent years, the simultaneous use of these two interesting features of ZnO has led to the rise of a new research field, the so-called piezo-photocatalysis [33,34,35]. In this contribution, the piezoelectric behaviour of ZnO microstructures obtained by means of four different aqueous solution based synthetic procedures, in wurtzite form, was tested, and the most responsive material, deposited using the layer-by-layer humid approach, was used to promote the piezo-photocatalysis of a methylene blue dye in an aqueous solution. The home-made designed device would represent a proof of concept for the use of a low cost (for both the materials synthesis and for the device fabrication), facile, sustainable and humid synthesis of ZnO piezo-photocatalytic microstructures for organic compounds photo-degradation in water [35]. In fact, this technology could find interesting applications, such as, for example, the simultaneous energy harvesting from wind (used as mechanical energy) and luminous energy from solar light for pollutants degradation. 

## 2. Materials and Methods

### 2.1. Synthesis of ZnO Crystals 

All the reagents (heptahydrate zinc sulphate, monohydrate citric acid, sodium hydroxide) were purchased from Merck (Darmstadt, Germany) and used without further purifications. A chemical precipitation was performed and, in particular, four procedures were developed (see Table 1). For example, in procedure A, 0.84 g of zinc sulphate and 1.5 g of citric acid were dissolved in a total volume of 60 mL of Milli-Q grade water. The mixture was kept under gentle stirring at room temperature (R.T.) until it became transparent. Then, 0.6 g of sodium hydroxide was added. The reaction mix was kept under stirring for 2 h and then sodium hydroxide of 0.4 g was performed. The solution was stirred overnight (O.N.) The obtained white precipitate was collected and rinsed with Milli-Q grade water by centrifugation (12,000 rpm for 10 min), for three times, and finally treated at 100 °C for 8 h in a static oven. 

### 2.2. Morphological and Spectroscopic Characterization of ZnO Samples 

The morphological characterization of ZnO crystals was performed by field emission scanning electron microscopy (FE-SEM) with a Zeiss Sigma microscope (Zeiss Sigma, Marly-le-Roi, France), operating in the range of 0–10 keV, and equipped with an in-lens secondary electron detector and an INCA energy dispersive spectroscopy (EDS) detector. Samples were mounted onto stainless-steel sample holders by double-sided carbon tape and grounded by silver paste. 

Rigaku D-Max/Ultima+ X-ray diffractometer (Rigaku, Monterotondo, Italy), with CuKα radiation (λ = 1.5406 Å), operating at 40 kV/30 mA with a step size of 0.02° in the Bragg geometry, was used to record XRD patterns.

The four ZnO obtained powders were characterized by Micro Raman Xplora (Horiba, Rome, Italy) with a laser at 532 nm (power 0.125 mW cm^−2^). The measurements were performed at room temperature. All samples were investigated in the spectral range between 300–1000 cm^–1^. UV-Vis spectroscopy measurements were performed by using a PerkinElmer Lambda 650 (Perkin-Elmer, Milan, Italy). FTIR spectroscopy investigations were performed by using a Spectrum One (PerkinElmer, Milan, Italy) instrumentation, equipped by an AmplifIR accessory distributed by SensIR Technologies for the multireflection analysis.

### 2.3. Piezoelectric Measurements 

First, 0.0236 g of ZnO obtained with the four developed procedures (A, B, C and D) was suspended in 1 mL of ethanol (96%, Merck, Kenilworth, NJ, USA) and deposited onto a flexible conductive, supported by the layer by layer (LbL) method, in order to obtain a mixed layer with octadecylamine (ODA) transferred from a chloroform solution (10^−5^ M). In particular, the solid support was vertically immersed in the ZnO suspension at a speed of 8 mm min^−1^, and the water substrate was evaporated at room temperature, waiting about 30 min. After this, the substrate was immersed in ODA chloroform solution (at a speed of 8 mm min^−1^), and a mixed layer ZnO/ODA was obtained. The procedure was repeated to obtain multilayer samples. Piezopotentials were induced, fixing the solid support covered by ZnO/ODA layers on a sample holder end, free to be mechanically bent. The strain was introduced by using a mechanical stress on the terminal part of the sample (a pressure of 10 KPa was applied) and the bias was measured by means of a Keithley 2000 multimeter (Tektronix, Beaverton, OR, USA)

### 2.4. Photo-Degradation Experiment in Aqueous Solution

Methylene blue was chosen as a model compound in this contribution. More in detail, 2 mg of the catalyst sample (ZnO crystals obtained by procedure A) was suspended in a 10 mL aqueous solution of MB (10 mg/L). The mixture was stirred at room temperature in the dark for 1 h to reach the adsorption–desorption equilibrium on the catalyst surface, before starting irradiation by means of A LOT-Oriel Solar S class A (AM 1) solar simulator, placed at a fixed distance of 10 cm. The reaction was carried out under continuous stirring for 2 h. At each time point, 2 mL were withdrawn and characterized by UV-Vis spectroscopy, in order to monitor the percentage of degradation of methylene blue in time, at the absorbance value of 670 nm, corresponding to the maximum value of absorbance in the investigated conditions, by the following equation:% degradation = (A_0_ − A_i_/A_0_) × 100
where A_0_ is the absorbance before the irradiation, whereas A_i_ is the absorbance at the time t, both measured at the maximum intensity wavelength. 

### 2.5. Piezo-Photo Degradation Measurements 

ZnO crystals obtained by procedure A (0.0236 g) were suspended in ethanol (1 mL) and deposited, in a mixed layer with ODA, onto a flexible conductive support (1 cm × 1 cm), by the LbL method.

Before starting, the obtained hybrid layer was immersed in Milli-Q grade water for 30 min, to permit the desorption of ZnO excess. Then, the obtained photo-active layer (ODA/ZnO multilayers) was immersed into a methylene blue (10 mg/L) aqueous solution kept hanging on a dipper, ensuring that the sample was held for the duration of the experiment. The sample was mechanically strained by using rotating small propeller blades at 4 Hz frequency (see Figure 1). The irradiation source, a LOT-Oriel Solar S class A (AM 1), was placed at a distance of 10 cm to the surface of the methylene blue solution.

The methylene blue degradation was tested in three conditions: in the dark under mechanical strain, under irradiation and under both mechanical strain and irradiation. Before starting the monitoring of the methylene blue UV-Vis absorption spectrum, performed to test the degradation performance, the solution was kept under gentle stirring for 1 h, in order to reach the adsorption-desorption equilibrium of the dye on the surface of the immersed active layer. Then, the degradation reaction was carried out for 2 h and, at each time point, 2 mL of the solution was withdrawn and analysed by means of UV-Vis spectroscopy.

## 3. Discussion

### 3.1. Synthesis and Morphological Characterization of ZnO Samples 

In order to obtain 1D elongated ZnO microstructures, an easy two-step synthetic procedure was ad hoc modified and optimized [30]. In detail, the first step envisaged zinc hydroxide precipitation by the addition of NaOH to an aqueous mixture of zinc sulphate, as precursor, and citric acid as a complexing and stabilizing agent, able to coordinate the Zn^2+^ species, controlling concomitantly the size and shape of the forming ZnO microstructures [36]. The second step was represented by a thermal treatment at 100 °C of 8 h to provide crystallization of the ZnO structures in the wurtzite form, thus guaranteeing the piezoelectric properties [30]. Moreover, the use of such a mild thermal treatment represents a strong advantage, because it avoids the degradation of the citrate capping ligand, which coordinates the surface of the ZnO microstructures, allowing post-synthesis dispersion in solvents and processing from solution, for different kinds of applications [37].

The synthetic route can strongly modulate the ZnO crystalline phase, size and morphology, depending on the chosen precursors and relative concentrations, reaction time, pH and temperature [38]. According to this, four different ZnO synthesis protocols (herein called A, B, C and D) were carried out in this contribution, and the obtained ZnO microstructures were morphologically characterized after the thermal treatment by means of scanning electron microscopy (SEM). 

The synthetic parameters and conditions of the four procedures are schematized in Table 1.

ZnSO_4_ was chosen as a precursor, since other Zn salts, such as Zn nitrate or Zn chloride, are reported to be not able to ensure the formation of elongated ZnO structures, by means of a chemical precipitation approach [39]. The concentrations of ZnSO_4_ and citric acid were kept constant in all the procedures, whilst the effect of a double addition of NaOH and of two different times of reaction elapsed after the first and the second addition of NaOH were investigated. 

In particular, the procedure A of Table 1 envisaged a first addition of NaOH, to the ZnSO_4_ and citric acid reaction solution. After stirring of the reaction mixture for 2 h, at room temperature (R.T.), a second aliquot of NaOH was added, and the reaction was further left to occur, under gentle stirring, overnight at R.T. After a thermal treatment at 100 °C in a static oven, the obtained ZnO powder was characterized by means of SEM (Figure 2A,B).

The secondary electron microscopy images show hexagonal columnar-like ZnO structures of different lengths and diameters; the former ranging between few hundreds nm to about 1 µm, the latter from few tens nm to some hundreds. 

When the synthesis was carried out following the procedure B (Table 1), that differs from procedure A in the time elapsed after the second addition of NaOH, 2 h instead of overnight, ZnO aggregated microstructures of ca. 1–2 µm were mostly obtained (Figure 3A), as well as a minority of hexagonal ZnO rods (white circle in Figure 3B). The results allow one to infer that the reaction time of 2 h, after the second addition of NaOH, is not sufficiently long to induce the formation of the columnar ZnO microstructures obtained in route A, upon thermal treatment.

This result prompted us to test the procedures C and D (Table 1), in order to understand if the directional growth was ruled just by the reaction time if all NaOH was added at the beginning (procedure C and D), or such a homolog growth was ensured by the second adjunct of NaOH and by the duration of the reaction after this, as envisaged in procedure A.

The ZnO microstructures obtained by following procedure C (Table 1) are characterized by a smooth, flake-like morphology (Figure 4A), assessing that, although all the amount of NaOH was added in a single injection to the zinc sulphate and citric acid reaction solution, the reaction time of 2 h is not sufficient to guarantee the growth of the elongated ZnO structures upon thermal treatment.

On the other hand, the hexagonal elongated ZnO microstructures were not synthesized neither by following procedure D, that relays on a single injection of all the amount of NaOH, as in route C, but differs from the time elapsed after such an addition, that is overnight instead of 2 h (Figure 4B). Basically, in this case, a few hexagonal structures of about 500 nm in length are detected, but removing the second NaOH addition, the duration reaction is not enough to reproduce the morphology obtained in procedure A. The produced ZnO structures are characterized by a hexagonal columnar shape with poorly defined faces. In this procedure, the control of the growth is reduced, suggesting the importance of both the two NaOH additions and the duration of the two steps of the zinc hydroxide precipitation. Furthermore, in detail, the experimental conditions, used in approach A in this contribution, probably ensured a mechanism based on the nucleation of zinc hydroxide seeds and then the preferential growth along the c-axis (001 plane). In particular, double columnar ZnO crystals are obtained, which are usually reported to be obtained under hydrothermal conditions, at a temperature between 100 °C and 300 °C and a pressure higher than the atmospheric one, in the presence of organic coordinating agents [40,41]. In the proposed procedure, in standard conditions, it can be supposed that Zn^2+^ ions, in strong alkaline conditions, are involved in the formation of Zn(OH)_2_ species, which are complexed by carboxylate groups of citrate during the first 2 h of reaction; the second addition of NaOH could convert part of the Zn(OH)_2_ into [Zn(OH)_4_]^2-^ in tetrahedral grown units [42,43,44,45], and could allow overcoming the supersaturation [46], permitting the second nucleation, and hence, the growth along the c-axis also due to the presence of citric acid [42]. The existing balance among seed nucleation rate, crystals growth rate and second nucleation rate in this way was shifted towards the second nucleation, slowing down the crystal growth rate and allowing the controlled and directional elongation of the crystals. The formation of the fuse with a half interface between two columnar crystals could be ascribed to the presence of the citrate capping agent in the reaction mixture [47], and of Na^+^ ions in solution, which could act as mineralizer binding two different grown units [48], with the consequent formation of a double crystal seed. The experimental conditions, then, drove the elongation in the two opposite directions. It is worth underlining that the developed approach allows one to tune the shape and chemical physical features of zinc oxide microstructures, avoiding expensive and sophisticated synthesis conditions.

### 3.2. Spectroscopic Characterization of ZnO Samples

The crystalline structures of the samples were characterized by means of X-ray diffraction. The diffraction patterns of the four samples are very similar, and they can be associated to the crystalline form of nanostructured ZnO (JPCS card 36-1451) in the wurtzite phase. In Figure 5, the XRD pattern of the samples A, B, C and D are reported, and the crystalline planes corresponding to the most relevant signals are labelled.

ZnO powders have been further characterized by means of Raman spectroscopy (Figure 6). The Raman technique is often used to characterize nanostructured ZnO, since it gives information both on the crystal phase of ZnO and on the presence of defects in the lattice structure. The main peak, located at about 440 cm^−1^ and present in all samples, is attributed to the so-called E_2_ (high) vibration, typical of the wurtzite form of ZnO [49]. Close to this characteristic peak, a signal at about 475 cm^−1^ can be observed in the samples B, C and D. As reported in the literature, it can be assigned to interfacial surface optical phonons, and its intensity strongly depends on the crystalline phase and synthesis procedure used to obtain the ZnO structures [50,51,52]. At 330 cm^−1^, the E_2_(high)-E_2_(low) vibration mode promotes the comparison of a Raman signal in all the examined samples [49]. The signal at 570 cm^−1^ is caused by the formation of structural faults, in particular oxygen and interstitial zinc atom vacancies [49]. In the case of sample A, this peak appears broader and less intense if compared to the other Raman spectra of samples B, C and D. Another interesting peculiarity that can be observed in the Raman spectra of the four samples is the position and intensity of the signal attributed to the A_1_(TO) vibration [49] (located at about 385 cm^−1^ for the sample A, 382 cm^−1^ for the samples B and C and 376 cm^−1^ for the sample D), that is influenced by the ZnO form, by defects and by the doping materials presence [53,54,55]. At high wavenumbers, about 845 cm^−1^, a broad band assigned to the A_1_(TO)+E_2L_ multiphonon scattering mode appears in the four samples.

### 3.3. Piezoelectric and Piezo-Photocatalytic Responses of the ZnO Films

The four samples were transferred onto a metallic plate using the LbL method, according to the procedure reported in the Materials and Methods section. The thickness of A, B, C and D samples, measured by means of a contact profilometer, are comparable (Appendix A). 

Mixed layers of ZnO and octadecylamine (ODA) were deposited according to the procedure reported in the Materials and Methods section. Since ZnO powder can be easily dispersed in aqueous solution, ODA molecules were used to anchor ZnO. Furthermore, ODA film was used to protect the ZnO transferred layer from the external mechanical solicitation.

An external mechanical deformation was applied by means of a pressure application of 10 KPa on the substrate (see the Materials and Methods section for the experimental details). All the samples showed the generation of a piezopotential after the application of the external mechanical stimulus, even though considerable differences can be observed for the four samples (Figure 7). For all of them, the piezoelectric responses do not seem to be influenced by the frequency of the applied mechanical deformation. In fact, the frequency of the cyclic pressure applied was changed from 1 to 3 Hz, and the potential, generated during the stress, does not significantly change. On the other side, the four examined samples showed interesting differences in the intensity of the generated bias. A piezopotential of about −0.1 V was obtained when sample A was subjected to an external pressure of 10 KPa. The sample B response is 40% lower than the bias recorded for the sample A, and the intensity of the recorded voltages decreases down to −0.055 V and −0.045 V for the samples C and D, respectively. 

A possible rationale to justify the observed differences is that two different morphologies can be identified in the synthetized samples [56]. Elongated structures can be identified in the sample A (hexagonal rod of ZnO); as is known, a charge separation is generated with an accumulation of the positive charges on the Zn-terminated plane and negative ones on the O-terminated crystalline plane along the c-axis [57]. The pseudospherical structures obtained by the synthesis B, C and D can represent a limit to an efficient charge separation following the mechanical stress [57].

Sample A, that appeared the most performing as piezoelectric substrate, was further characterized. The piezoelectric response was evaluated as a function of the ZnO deposited film thickness. Four samples of different thicknesses were deposited (Appendix A) and the piezoelectric responses were evaluated (Appendix A). An asymptotic behaviour was observed and a plateau was reached when 10 double layers of ODA/ZnO were transferred (about 0.34 μm). 

Sample A was used to induce the photo-degradation of methylene blue dye in solution. A LOT-Oriel Solar S class A (AM 1) (LOT-Quantum Design, Rome, Itay) solar simulator was used to promote the formation of electron (e^−^)/hole (h^+^) in the ZnO crystals. As reported in Figure 8, the maximum absorption of sample A is centred at 370 nm, so that relevant part of emitted photons can be absorbed by the ZnO crystals. 

In Appendix A, the bleaching of the maximum absorption peak of methylene blue confirms, as known in the literature [58], the photo-induced degradation of the dye. In particular, ZnO crystals of sample A (0.2 g/L) were mixed with the dye (0.01 g/L) in Milli-Q grade water, and the suspension was kept under continuous and gentle stirring and illuminated for 2 h. The maximum absorption band of methylene blue is located at about 670 nm and, at each time point, its photo-degradation was monitored by means of UV-Vis spectroscopy. The possibility to re-use the catalyst was explored. After 60 min under irradiation in the presence of the dye, the catalyst was recovered by means of centrifugation and re-utilized three times. After the first cycle, the percentage of degradation was about 40% in the tested experimental conditions. Such value was reduced in the next three cycle of re-utilization down to ca 35%, confirming the possibility of re-using the catalyst, in agreement with the literature [32]. 

The next step was represented by the investigation about the piezo-photocatalytic activity of sample A, deposited onto the solid support. Before performing such characterization, the protecting role of the ODA layer from the ZnO water re-suspension was evaluated by monitoring the absorption peak of ZnO. In particular, 5 layers of ODA/ZnO mixed layers were transferred by the LbL method on a quartz slide, and the absorption spectrum was registered (dotted line in Appendix A). Then, the solid support was immersed in water and the absorption spectra were recorded each 10 min (continuous line). A bleaching of the absorption peak is observed after the first immersion, then the amount of ZnO present on the solid support appears almost unchanged, confirming the protective role of ODA. The interaction between the organic and the inorganic layer was investigated by FTIR in multireflection mode (Appendix A). In the region 4000–3000 cm^−1^, asymmetric and symmetric stretching modes of primary ODA amine are located at 3330 cm^−1^ and 3255 cm^−1^, respectively. Both bands appeared shifted if compared with the same vibrational modes of the ODA when deposited on the ZnO layer. This evidence suggests that ZnO is chelated by the primary amine group of ODA [42].

The piezo enhanced photo-degradation of methylene blue was obtained using two simultaneous external triggers: the mechanical stimulus, that produces a charge accumulation on the ZnO surfaces, and the UV-Vis light. This phenomenon, known as photo-piezo degradation [59,60,61], is a relatively recent approach to induce the catalysis of organic compounds [33]. The piezo-enhanced photo-induced degradation mechanism can be schematized as in Figure 1: UV photons promote the electrons in the ZnO conduction band and an equal number of holes in the valence band (Figure 9A). The deformation of the ZnO crystal induces the formation of an electric field with the negative pole on the compressed face (in red in Figure 9B) and the positive one on the opposite face (in red in Figure 9B) [56]. The electrons will be more efficiently separated by the presence of the external electric field, reducing the charges’ recombination. Furthermore, the localization of holes and electrons on the surfaces of ZnO enhances the degradation reactions (Figure 9C). The home-made experimental layout is sketched in Figure 1: the substrate, covered by the sample A/ODA thin film, is immersed in a methylene blue water solution, and an external mechanical stress is cyclically applied, with the possibility to modulate the frequency of the mechanic stimulus. Simultaneously, the substrate is irradiated by means of the light source, ensuring that the beam does not intercept the backer walls. 

In Figure 10, the degradation rates of methylene blue in water solution, upon illumination, by applying a mechanical deformation of the substrate and under both the external stimuli, are compared. 

As is easy to predict, the mechanical stimulus does not produce any relevant difference in the absorption intensity of methylene blue solution. It is interesting to observe that, instead, when the mechanical stimulus is used to promote the formation of the piezo-potential, a relevant enhancement of the photo-degradation rate of methylene blue is recorded, confirming the proposed mechanism. For the sake of clarity, the degradation efficiency proposed in the present proof of concept is not particularly high, even though it has to be stressed that the experimental conditions during the piezo-photo experiments are particularly limiting: in a common photo-degradation experiment, the photocatalyst is used as a suspension in the aqueous solution of the dye and the colloidal suspension is continuously stirred. On the other side, the possibility to four times increase the degradation efficiency using, for example, the completely green and renewable energy, such as wind or sea waves, is very appealing. 

## 4. Conclusions

Four different synthetic protocols were used to obtain ZnO microstructures. Mild conditions allowed to preserve the organic capping agent obtaining elongated well-defined hexagonal ZnO microstructures, with features usually achieved using more complex and expensive methods. The synthesized samples have been characterized by means of Raman and XRD spectroscopy, which evidence that all the samples are ZnO in the wurtzite phase, even though spectroscopic differences have been highlighted, as a consequence of the presence of vacancies and defects in the crystalline structures. Strong morphological differences on a microscale have been shown up by SEM analysis, that evidenced at least three different morphologies: flake-like, spherical and elongated hexagonal structures. ZnO films were transferred onto different supports by the layer by layer method, and films’ thickness was evaluated in order to ensure the control and repeatability of the transfer method. The piezoelectric responses of the four samples have been measured and, as predicted from the spectroscopic and morphological results, the hexagonal ZnO microstructures show the higher piezo induced bias. Again, the photo-degradation characteristics of the most performing ZnO sample have been checked on water dissolved methylene blue dye. Finally, the photo-degradation process has been improved, inducing the piezoelectric bias on solid ZnO film by means of a mechanical stress, and about 10% of the degradation efficiency was recorded.

## Figures and Tables

**Figure 1 materials-13-02938-f001:**
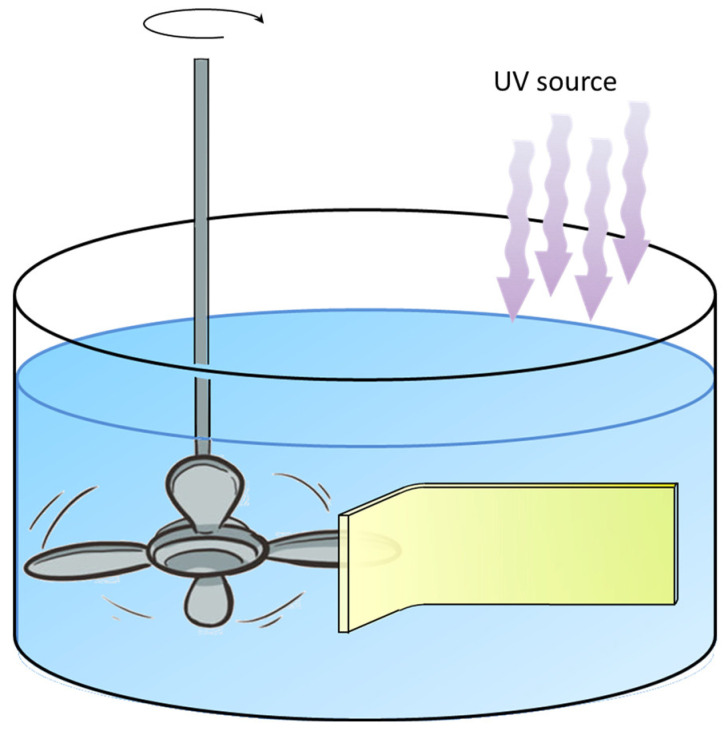
The home-made experimental layout is represented: the substrate covered by the piezoelectric film is immersed in the dye aqueous solution containing and it is irradiated by means of an UV source. Simultaneously, the solid support is mechanically stressed by means of using rotating small propeller blades at 4 Hz frequency.

**Figure 2 materials-13-02938-f002:**
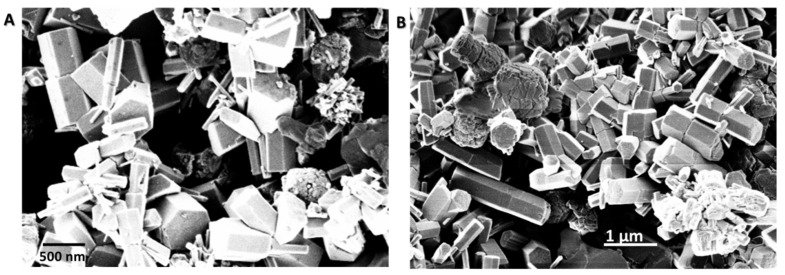
SEM images of ZnO rod-like microstructures synthesized by following procedure A: in panel (**A**) the image was acquiredat 30 Kx magnifications, in panel (**B**) 20 Kx magnifications were used.

**Figure 3 materials-13-02938-f003:**
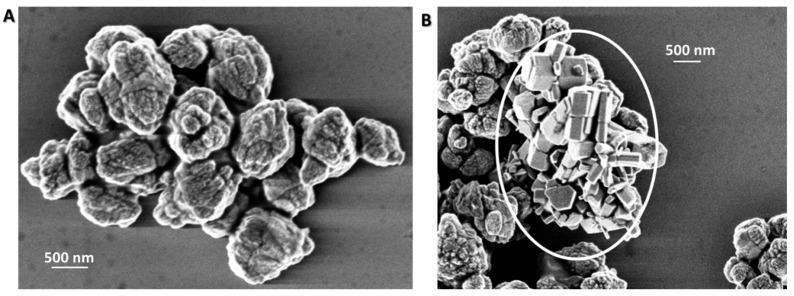
SEM images acquired in two different regions of the same ZnO sample obtained by procedure B after the thermal treatment (30 Kx). In particular, in panel (**A**) aggregated microstructures are visible, in panel (**B**) hexagonal ZnO rods are present.

**Figure 4 materials-13-02938-f004:**
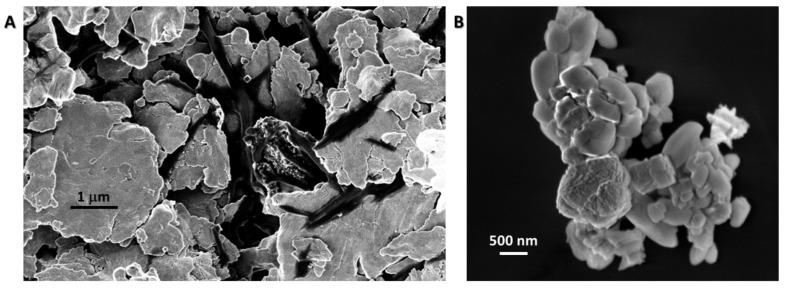
SEM images of ZnO microstructures synthesized by following procedure C (panel **A**) and D (panel **B**) (magnification 30 Kx).

**Figure 5 materials-13-02938-f005:**
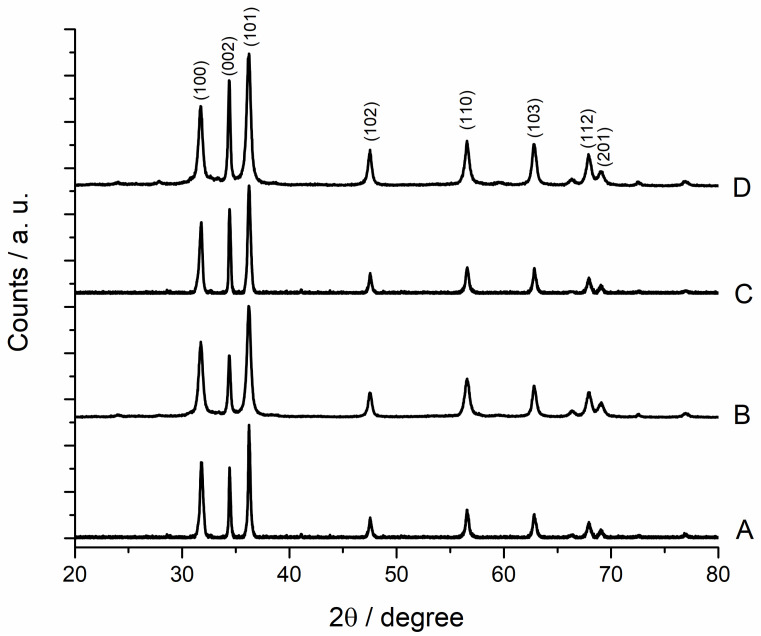
XRD patterns of ZnO powder of the samples A, B, C and D. In the graph, the crystalline planes responsible for the most relevant diffraction peaks are reported.

**Figure 6 materials-13-02938-f006:**
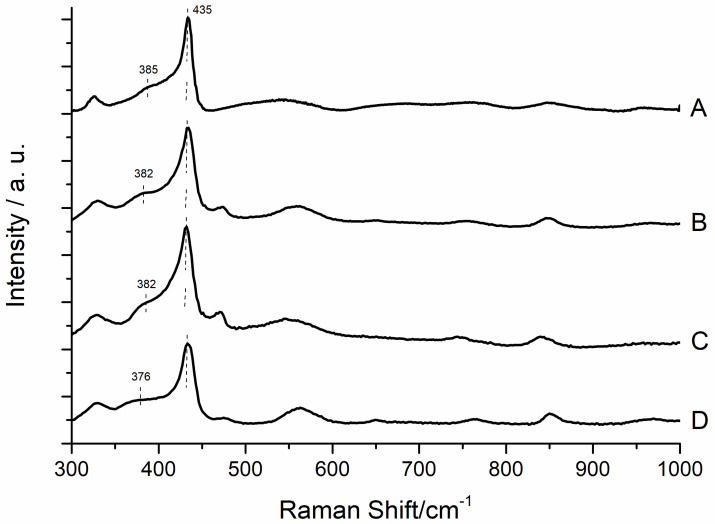
Raman spectra of the four ZnO sample powders in the range 300–1000 cm^−1^.

**Figure 7 materials-13-02938-f007:**
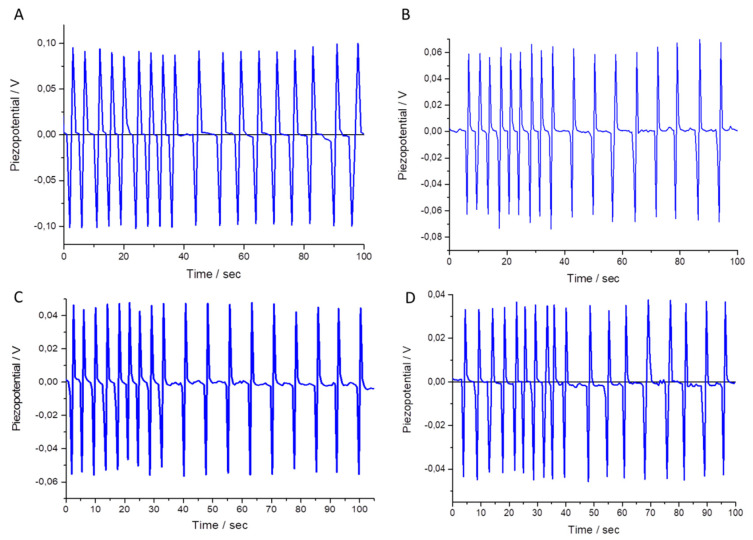
Generation of a bias by means of the application of a cyclic external pressure on the piezoelectric thin films **A**, **B**, **C** and **D**. Two different frequencies for applying the mechanical stimulus were used (3 Hz and 1 Hz). For all samples, after the negative bias generated by the stress application a positive V is recorded during the release phase. All the curves are shown as the average of different measurements carried out on five different films for each kind of sample.

**Figure 8 materials-13-02938-f008:**
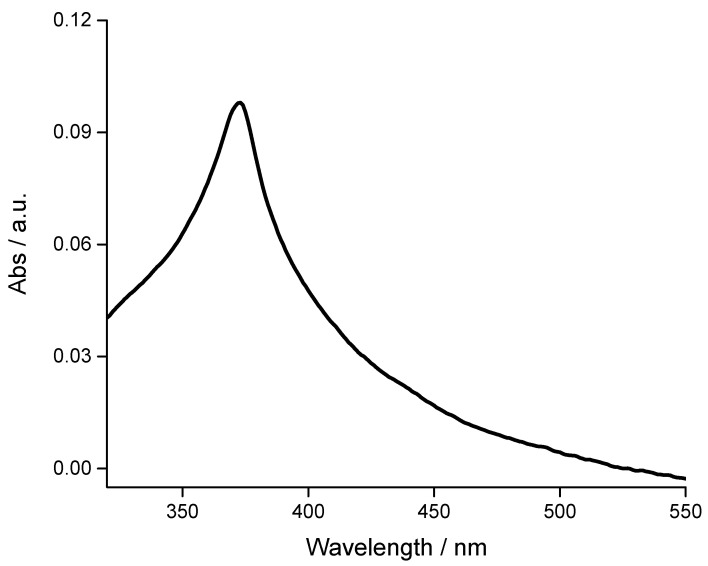
Absorption spectrum of ZnO crystal (sample A) suspension in ethanol in the wavelength range 320–550 nm.

**Figure 9 materials-13-02938-f009:**
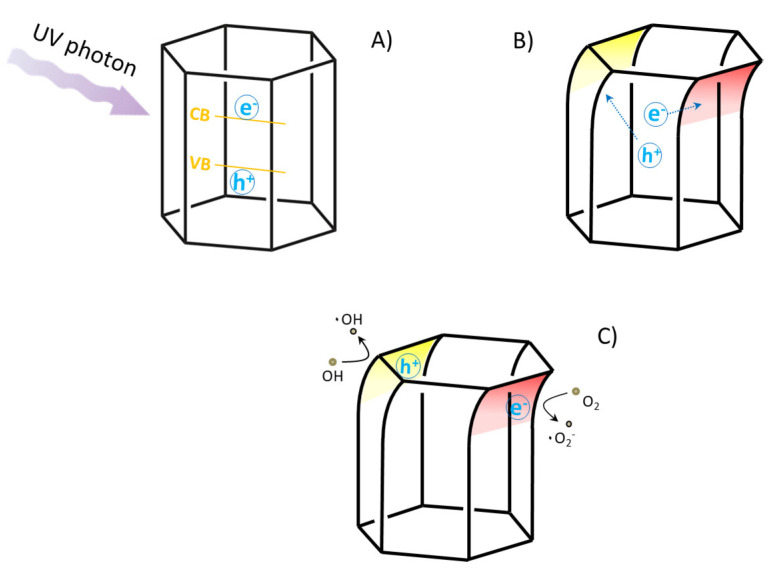
(**A**) The electron is promoted from the valence band to the conduction band of ZnO by means of the appropriate illumination. (**B**) The hole/electron couple is separated by the presence of the piezo-electric field, and the separated charges migrate towards the ZnO surfaces, where the degradation reactions can take place (**C**).

**Figure 10 materials-13-02938-f010:**
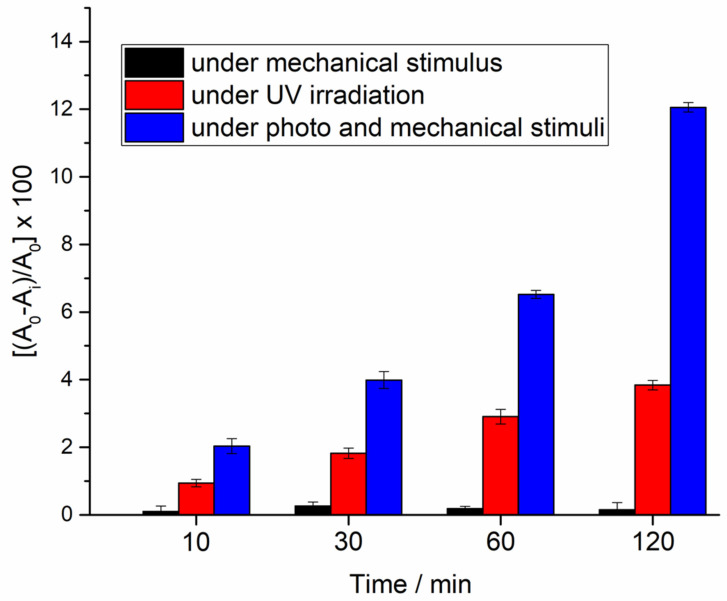
Comparison of the percentage of degradation of the methylene blue recorded at different laps of time under photo, piezo and piezo-photo stimuli.

**Table 1 materials-13-02938-t001:** Zinc oxide synthesis procedures.

	ZnSO_4_^.^ 7H_2_O/g	Citric Acid/g	NaOH (1st add)/g	Milli-Q Water/mL	Stirring @R.T.	NaOH (2nd add)/g	Stirring @R.T.	Thermal Treatment @100 °C
A	0.84	1.5	0.6	60	2 h	0.4	overnight	8 h
B	0.84	1.5	0.6	60	2 h	0.4	2 h	8 h
C	0.84	1.5	1	60	2 h	/	/	8 h
D	0.84	1.5	1	60	overnight	/	/	8 h

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
