# Peer review of "Wet Synthesis of Elongated Hexagonal ZnO Microstructures for Applications as Photo-Piezoelectric Catalysts"

_materials, 2020, doi:10.3390/ma13132938_

Round 1

Reviewer 1 Report

In the reviewed manuscript, the growth of elongated hexagonal ZnO microparticles for photo-piezoelectric catalysts has been studied. The manuscript is well written with a few minor shortcomings mentioned below.

1. In the synthesis part, the following statement has been made:

"The concentrations of ZnSO4, citric acid and NaOH were kept constant
89 in all the procedures, whilst the effect of a double addition of NaOH"

From Table 1, we clearly see that the initial concentration of NaOH is different which is contradicting to the above. Please reconsider the description to avoid confusion.

2. Selected peaks of Raman spectra were discussed avoiding suggestions for mechanisms of the peaks around 330, 470 and 850 cm-1.

3. It is not clear how many thin film samples for each technique A-D were tested to get a piezoelectric response from 0.1 to 0.4V. Please provide information about the statistical analysis of the data obtained.

4. The use of punctuation, spaces, some grammar have to be updated to the professional standard of the peer-reviewed scientific journal.

Reviewer 2 Report

The authors introduce a two-step injection process to synthesize ZnO with rod-like microstructure. The rod-like microstructure can effectively enhance the charge separation and achieve a twofold increase of piezoelectric response. The photodegradation experiments are carried out by the best performing ZnO, and a fourfold increase of photodegradation efficiency is reported under the synergic effect of photon and mechanical stimuli. I think this study will provide interest to the general readership of this journal after some experiments being done to fill in the missing details.

1. About the experimental design, the author compared the two-step injection strategy (protocol A and B) with the one-pot synthesis process (protocol C and D) to prepare ZnO microstructure while keeping the overall amount of NaOH a constant. However, the author did not explain the specific dosage of NaOH in each injection for a two-step injection strategy. Can the authors discuss the choice of 0.6/0.4 dosages in the two-step injection process instead of other combinations? Now that it is of importance to control the injection sequence of NaOH, it would definitely bring more insight if the author can carry out experiments with different dosage combinations, such as 0.2/0.8, 0.4/0.6, 0.8/0.2, and study the results.

2. The authors design four synthesis protocols to prepare ZnO. However, the protocols end up with a mixture containing an equal amount of chemicals and only distinguish from each other by the sequence of mixing before thermal treatment. Can the author provide more explanation about how the sequence of mixing influences the final morphology?

3. In Figure 5, why the Raman shift of sample D (380 to 376 cm-1, redshift) is different from the other samples (380 to 382 cm-1, blue shift)?

Reviewer 3 Report

Authors spend a lot of effort to discuss the growth of ZnO nanostructure with the hexagonal shaped ZnO rods/wires as the targeting product. But the results as catalogues as A, B, C D products are not charming.There have a lot of reports and review work on ZnO nanostructures growth with well controlled facets and morphologies in the past twenty years. Therefore these growth work reported here cannot guarantee the acceptance of the manuscripts.

Besides what I mentioned above, why authors need the elongated hexagonal ZnO microstructures for applications as photo piezoelectric catalysts? The devices authors fabricated are thin films. 

Even the mechanism is well known. therefore, there is no fresh idea or results reported in this work.
